# Estimating the Mass of Galactic Components Using Machine Learning Algorithms

Jessica N. López-Sánchez [1,2,]*, Erick Munive-Villa [1,2], Ana A. Avilez-López [1] and Oscar M. Martínez-Bravo [1]

1 CEICO—FZU, Institute of Physics of the Czech Academy of Sciences, Na Slovance 1999/2, 182 00 Prague, Czech Republic; munive@fzu.cz (E.M.-V.); aavilez@fcfm.buap.mx (A.A.A.-L.); omartin@fcfm.buap.mx (O.M.M.-B.)
2 Facultad de Ciencias Físico-Matemáticas, Ciudad Universitaria, Benemérita Universidad Autónoma de Puebla, Av. San Claudio SN, Col. San Manuel, Puebla 72592, Mexico
* Correspondence: lopez@fzu.cz

**Abstract:** The estimation of galactic component masses can be carried out through various approaches that involve a host of assumptions about baryon dynamics or the dark matter model. In contrast, this work introduces an alternative method for predicting the masses of the disk, bulge, stellar, and total mass using the k-nearest neighbours, linear regression, random forest, and neural network (NN) algorithms, reducing the dependence on any particular hypothesis. The ugriz photometric system was selected as the set of input features, and the training was performed using spiral galaxies in Guo's mock catalogue from the Millennium simulation. In general, all of the algorithms provide good predictions for the galaxy's mass from $10^9\ M_\odot$ to $10^{11}\ M_\odot$, corresponding to the central region of the training domain. The NN algorithm showed the best performance. To validate the algorithm, we used the SDSS survey and found that the predictions of disk-dominant galaxies' masses lie within a 99% confidence level, while galaxies with larger bulges are predicted at a 95% confidence level. The NN also reveals scaling relations between mass components and magnitudes. However, predictions for less luminous galaxies are biased due to observational limitations. Our study demonstrates the efficacy of these methods with the potential for further enhancement through the addition of observational data or galactic dynamics.

**Keywords:** galactic systems; neural network; scaling relations

## 1. Introduction

The bulge–disk decomposition of galactic systems is useful for understanding the evolutionary processes of galaxies. Specifically, the disk and bulge masses can be inferred, given that their stellar population has different dynamic or even chemical features. There are plenty of schemes for classifying galaxies; one of the most popular corresponds to the morphological classification proposed by Edwin Hubble [1], which distinguishes four different types of galaxies: elliptical, spiral, barred spiral, and irregular. Another method involves the isophotal radius measurement [2], determining the size attributed to a galaxy component according to the corresponding surface brightness level. A way to characterise the light distribution independent of the light profile is through the concentration measure, defined by the ratio of two geometrical regions, each containing a fixed fraction of the total luminosity of the galactic system [3].

Another approach for reconstructing the visible mass of galactic components involves using standardised fitting functions. Ideally, these functions should be derived from the fundamental principles governing galactic evolution. However, due to the intricate nature of the physics involved, models based on these principles often become complex, with a substantial number of parameters. Then, commonly used functions are empirically derived. For instance, the disk components are well-fitted by an exponential law, while for the elliptical galaxies and the bulges in the spiral ones, the relations that are typically considered

are King's model [4] and de Vaucouleurs's law [5]. Sometimes, the bulges associated with late-type galaxies are best fitted by exponential laws [6,7]. However, implementing these methods demands high-quality observational data to obtain reliable results.

Furthermore, it is possible to single out the galactic components through the light distribution of a galaxy. This decomposition is derived by fitting the light profile to a power law, adhering to a specific empirical or analytical model. In the conventional photometric technique, the one-dimensional case is considered. On the other hand, when multiple wavelengths in the spectrum are taken, spectroscopic methods come into play [8].

On the other side, numerical simulations play an important role in exploring predictions of galaxy evolution within the standard $\Lambda$CDM prescription [9–11]. Semi-analytical models have gained popularity when identifying the structural components of galactic systems. These models employ a simplified representation of baryonic physics, coupled with Markov chain Monte Carlo methods for reconstructing merger trees [12].

For dark matter-only simulations, a common technique to infer information about the baryonic components is to assume the *halo abundance matching* (HAM), which relates the halo potential well to the star formation rate in such a way that more luminous galaxies are associated with more massive halos. During the evolution of both components, material exchange occurs between the baryonic elements through various processes. For example, forming a galactic bulge may result from major or minor mergers [13]. In these processes, pre-existing and newly formed stars play a crucial role; after a merger, stars from the progenitors contribute to the bulge component of the resulting galaxy. The gas within the progenitors becomes part of the resulting galaxy disk, and the specific angular momentum of this component equals that of the halo in which it is embedded [14–16].

As can be seen, various approaches exist for describing galactic components, including purely morphological observations or photometric and/or spectroscopic techniques, either being synthetic catalogues. Conversely, obtaining information about total mass often involves making strong assumptions about either a specific dark matter model or about the overall kinematics of the system. In this sense, it has been shown that ML methods can reduce the dependence on such assumptions [17]. For example, in [18], the authors propose a random forest-based approach to predict the total and dark matter masses of galaxies using simple observations from photometric and spectroscopic studies, while [19] presents a supervised ML method to display multidimensional information on stellar populations and kinematics in the MaNGA study in a 2D plane. Additionally, in [17], a sample of galaxies from the Illustris TNG simulation was used to predict the stellar and total masses using a convolutional neural network.

In this work, we propose an artificial intelligence (AI)-based method to isolate the bulge and disk components of both baryonic and total galaxy mass. This is accomplished using the information on luminosity and features inferred from stellar dynamics encrypted in Guo's synthetic catalogue [14]. Our goal is to perform the decomposition of the galactic components without including additional information about baryons in the training stage beyond the patterns the AI methods can infer from the catalogue[1]. This method can be useful to predict the mass of the components of observed galaxies whose baryonic dynamics cannot be easily obtained using conventional techniques.

In this context, we will consider the following components: the bulge, the disk, and the stellar and total mass. Because the mass values range between several orders of magnitude, it is well suited for predicting the logarithm base 10. Here, the bulge mass $M_{\text{bulge}}$ is computed in terms of the disk mass $M_{\text{disk}}$ and total mass $M_\star$ from the following expression[2] [12,14]

$$M_\star = M_{\text{bulge}} + M_{\text{disk}}. \tag{1}$$

We are interested in the strengths and weaknesses of the machine learning (ML) algorithms when keeping the features set as simple as possible. For that reason, we proposed as input data the photometric information in different colour bands and the maximum rotational velocity of the halo, constructed directly from the simulation and independently of baryonic dynamics[3]. We will compute an estimation of the percent error of the pre-

dictions given by the AI methods with respect to the actual value of the simulated data. Additionally, the SDSS database [20] will be considered to assess how well the predictions match observations. This will allow us to identify the ranges of luminosity and mass where the algorithms show the best accuracy and explore the properties of the corresponding galaxies. This analysis is particularly important if we want to implement this method in other surveys.

This paper is structured as follows: Section 2 presents and analyses the content of Guo's galaxy catalogue to determine the correlation between input features and output predictions, emphasising their importance during the training stage. Following this, Section 3 introduces the ML algorithms considered in this work and explores their dependency on variations in different parameters. Subsequently, in Section 4, we analyse the performance of each algorithm and the percent error of the predictions. Then, in Section 5, we apply the trained methods to predict masses of components in observed galaxies from the Sloan Digital Sky Survey [20] database. We derive various scaling relations commonly studied in the literature and identify the regions where the predictions are more accurate. Finally, we draw some conclusions in Section 6.

## 2. The Data

To train our ML algorithms, we have used Guo's galaxy catalogue [14] derived from the Millennium simulation, selecting only galaxies with non-zero bulges or disk components, leading to a set of 833,491 galaxies. This dataset was split randomly, assigning a common selection where 75% of total data was defined for training and 25% was defined to evaluate the performance of the algorithms. The Millenium simulation is a dark matter-only simulation, carried out under the $\Lambda$CDM prescription [21] using a customised version of the Gadget 2 code [22], with $2160^3$ particles within a box of $L = 500$ Mpc/h. This catalogue provides information about the merger history of each halo and the baryon content, split into five components: the stellar bulge, the stellar disk, a gas disk, a halo, a black hole, and an ejecta reservoir [10].

The analytical model implemented in Guo's catalogue considers that galaxies form within the central region of dark matter halos. The fitting function, which describes the average baryon fraction of a halo given the total mass, can be written in terms of its mass and redshifts [23]

$$f_b(z, M_{\text{tot}}) = f_b^{\text{cos}} \left( 1 + (2^{\alpha/3} - 1) \left[ \frac{M_{\text{tot}}}{M_c(z)} \right]^{-\alpha} \right)^{-3/\alpha}, \tag{2}$$

where the universal baryon fraction is usually taken as $f_b^{\text{cos}} = \dfrac{\Omega_b}{\Omega_0} \sim 17\%$. Here, $M_c$ represents the characteristic mass objects, which can retain 50% of the gas components to form stars. The reionisation and cooling depend on the baryon fraction in a given halo and on its mass and redshift. The disk and bulge formation are correlated with star formation and supernova feedback processes, as well as with the black hole growth and AGN feedback. Additionally, mergers between the central and satellite galaxies are described through simulations and play an important role in the disk and bulge evolution. This catalogue accurately reproduces the population and clustering mechanisms observed at $z \sim 0$. However, it exhibits inconsistencies for high-redshift populations.

In this work, we consider galaxies at $z = 0$. Our goal was to investigate spiral galaxies hosting both bulges and disks. Then, we imposed this strong filter when selecting our sample from the mock catalogue. It is crucial to note that our selection encompasses diverse galaxy types without accounting for age or metallicity. The purpose is to explore the capabilities of the algorithms to get information about the systems by exclusively using photometric information.

The resolution of the simulation delimits the range of masses for each component. Once the selection of bulge–disk galaxies has been performed, the range of the total mass

is between $10^{10} M_\odot/h$ and $10^{13} M_\odot/h$. Notably, the selected total mass range excludes galaxies of both low and high masses. Indeed, massive galaxies tend to exhibit an elliptical morphology rather than spiral [24].

*Features Importance*

It is well known that the physical and photometric properties of the stellar population of a galaxy are closely related to its dynamics and the spatial mass distribution of different components within the system. Specifically, this behaviour is reflected in the colour–magnitude relation. For instance, it has been shown that bulge-dominant galaxies have a color–magnitude diagram mainly described by red galaxies [25]. Additionally, in [26], it has been shown that the bulge is redder than the disk in galaxies within a cluster. A similar conclusion was reported in [27].

In ML, the training data are defined as the feature vector $\mathbf{X} = (\mathbf{X}_1, \mathbf{X}_2, \ldots, \mathbf{X}_n)$ and their corresponding label or associated output $y = (y_1, y_2, \ldots, yn)$, where $n$ is the sample size, with unknown distribution $\mathcal{P}(\mathbf{X}, y)$ as follows

$$D = \left\{ (\mathbf{X}_1, y_1), \ldots, (\mathbf{X}_n, y_n) \right\} \subseteq \mathcal{R}^d \times \mathcal{I}, \tag{3}$$

here $\mathcal{R}^d$ denotes the $d-$dimensional feature space and $\mathcal{I}$ is the label space. In this work, we consider two sets of features, $\mathbf{X}_I$ and $\mathbf{X}_{II}$. The first corresponds to the absolute magnitudes $\mathbf{X}_I = (u, g, r, i, z)$, which we hereafter refer to as Set I. Such magnitudes are also available in the SDSS dataset; therefore, there is an observational counterpart. Within a second set (Set II), the same features as Set I are considered in addition to the maximum rotational velocity of the halo $\mathbf{X}_{II} = (u, g, r, i, z, V_{\max})$. In both cases, the predictions (labels) are $y = (M_{\mathrm{disk}}, M_\star, M_{\mathrm{tot}})$, as it is displayed in Table 1. An exploration of the data for each set was conducted using Pearson's correlation ratio,

$$r_{\mathbf{X}, y} = \frac{\sum_{i=1}^n (\mathbf{X}_i - \bar{\mathbf{X}})(y_i - \bar{y})}{(n-1) S_X S_y}, \tag{4}$$

where barred symbols represent the mean values and $S_{X,y}$ is the standard deviation. When this quotient is $r_{X,y} = \pm 1$, we have a perfect positive (negative) correlation, whereas for $r_{X,y} = 0$, the parameters are not correlated at all.

**Table 1.** Input and output features considered for the ML algorithms in this work. Set I corresponds to the photometric information derived from Guo's catalogue using semi-analytical models. Set II includes information about the dynamics of all components.

| Input | $u$ | $r$ | $g$ | $i$ | $z$ | $V_{\mathbf{max}}$ |
|---|---|---|---|---|---|---|
| Set I ($\mathbf{X}_I$) | ✓ | ✓ | ✓ | ✓ | ✓ | |
| Set II ($\mathbf{X}_{II}$) | ✓ | ✓ | ✓ | ✓ | ✓ | ✓ |
| Output ($y$) | | $\log_{10}(M_{\mathrm{disk}})$ | $\log_{10}(M_\star)$ | $\log_{10}(M_{\mathrm{tot}})$ | | |

In Figure 1, the correlation matrix illustrates how the features contribute to the algorithm's predictions. For completeness, the mass of the bulge and the mass of the central black hole $M_{\mathrm{bh}}$ have been included to analyse the whole set of masses of the mock catalogue. The matrix displays the absolute values of Pearson's correlation ratio, focusing solely on the strength of the correlation parameter. As anticipated, $M_\star$ exhibits a high correlation with the magnitudes, particularly with the z and i bands, corresponding to the infrared and near-infrared regions of the spectrum, respectively. Observationally, the determination of luminous mass is significantly influenced by dust, with emissions in the optical band experiencing reddening. Conversely, this effect is negligible in the near infrared [28]. On the other hand, $M_{\mathrm{disk}}$ shows less correlation with the magnitudes compared with $M_\star$ and exhibits a weak relation with the remaining quantities. Furthermore, $M_{\mathrm{tot}}$ exhibits a strong

correlation with $M_\star$ due to the HAM relation implemented in the mock catalogues. The correlation between $M_{\text{tot}}$ and $v_{\text{max}}$, which encapsulates information about the dynamics of all components, surpasses that of other masses. Notably, the correlations with $M_{\text{disk}}$ are weak, suggesting that the dark matter component influences the stellar component as a whole rather than each component individually. Regarding $M_{\text{bulge}}$, this quantity is less correlated with the magnitudes than $M_{\text{disk}}$; instead, it is more linked to $M_{\text{bh}}$. This last one also shows a strong correlation with $M_{\text{tot}}$ and a weak link with the rest of the features. From here, we define our features as the more correlated ones, that is, the magnitudes, the masses $M_{\text{disk}}$, $M_\star$, $M_{\text{tot}}$ and, additionally, $v_{\text{max}}$, in agreement with Table 1.

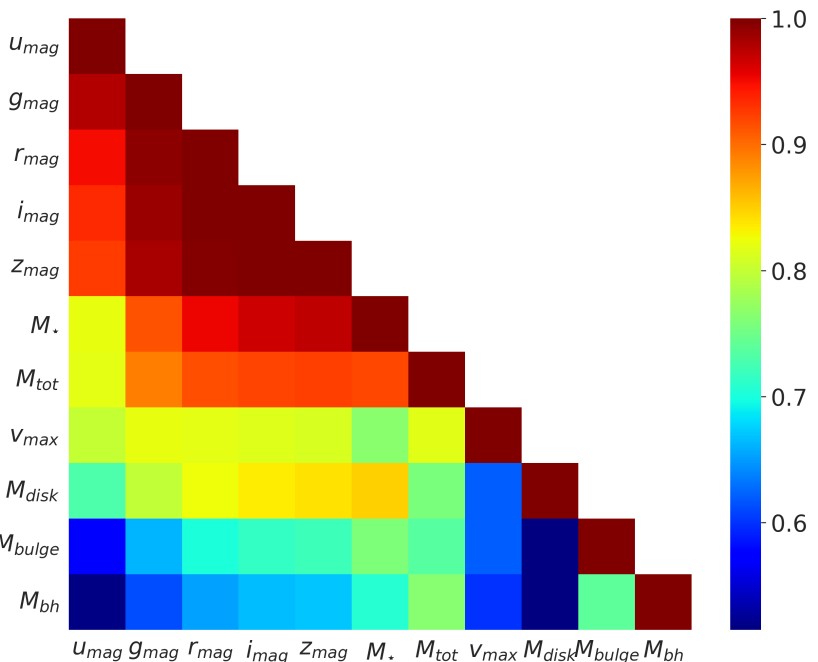

**Figure 1.** Heat map of the absolute value of the Pearson correlation coefficient between the galaxy parameters presented in Guo's catalogue. The redder the square, the higher the correlation. As expected, stronger correlations occur between different bands' stellar mass and magnitudes. However, a relation exists between the total mass and the magnitudes, although to a lesser extent.

## 3. Implementation of the ML Algorithms

We employed a set of widely used supervised algorithms that are known for their effective predictions, listed as follows. These methods were implemented using the scikit-learn library [29,30] and the Keras API [31].

- KN-Neighbours (KNN). This algorithm relies on the idea that the set of $k$ nearest data points $C_x \subset D$, where $|C_x| = k$ have similar values among them; here, $D$ is defined in Equation (3). To consider two points as neighbours, they should fulfill

$$\text{dist}(\mathbf{x}, \mathbf{x}') \geq \text{dist}_{\text{max}}(\mathbf{x}, \mathbf{x}''), \quad \text{with} \quad (\mathbf{x}'', y'') \in C_x, \quad (\mathbf{x}', y') \in D. \tag{5}$$

  This distance is defined in the hyperspace of features using the Euclidian metric, and the final value is the average of their outputs. In this case, the number of neighbours is a free parameter, and we found that the highest accuracy is achieved when the number of neighbours is close to 18; the error starts to increase beyond that value.
- Linear regression (LR). The traditional linear regression minimises the sum of the squared differences between the predicted and actual values. We are considering this method to compare it with more sophisticated techniques.
- Random forest (RF). This algorithm is subject to the number of trees and their depth. Each tree contains decision nodes $\mathcal{N}_m$ that split the data $(X_{\text{node}}, y_{\text{node}})$ (in the parent

node) into smaller (left and right) subsets in new child nodes $C_m^L$ and $C_m^R$ until the branch finds a homogeneous group according to the set of hyperparameters. Splitting each node in regression is conducted following the minimisation of the residual $\mathcal{R}_m$ as

$$\mathrm{argmin}(\mathcal{R}_m) = \sum_{m \in \mathcal{N}_m} (y_m - \bar{y}_m)^2 - \left( \sum_{m \in C_m^L} (y_- \bar{y}_m^L)^2 + \sum_{m \in C_m^R} (y_- \bar{y}_m^R)^2 \right), \qquad (6)$$

where $\bar{y}_m^L$ and $\bar{y}_m^R$ are the mean of the target values in the child nodes. In this case, the algorithm identifies patterns in the masses of the galaxies, and it is capable of categorising them into groups $y_m$ where certain requirements between the luminosities are fulfilled. The split is performed if the minimum $\mathcal{R}_m$ is below a defined threshold. Because of how the trees are built, it is easy to overfit. Therefore, it is strongly recommended to use a set of trees instead. We used nearly 150 trees for the training. The minimum number of samples required for splitting was four; below this number, the branches reach the maximum depth and are considered as pure.

- Neural network (NN). NN is an interconnected group of nodes stored in a layer and is connected to other nodes in the network by unidirectional connections of different weights. Patterns learned in a layer are transferred to the next activated nodes. We implement the early stopping-based method as a regularisation technique to avoid overfitting, stopping the training once the performance no longer improves. This is measured by the loss function, which quantifies the discrepancy between the predicted error and true values. For a regression, it can be taken as the squared loss function

$$\mathcal{L} = \frac{1}{n} \sum_{i=1}^{n} \left( h(\mathbf{X}_i) - y_i \right)^2, \qquad (7)$$

here, $h(\mathbf{X})$ is the function that minimises the loss associated with the target value of the $i$-th class, $h = \mathrm{argmin}(\mathcal{L})$. A common assumption is to take $h(\mathbf{x}) = \mathbf{B}^T \mathbf{X}_i + b$, where instances of $B$ are considered as the weights coefficients and $b$ is a constant. In this case, we also considered the Lasso regularisation method. This technique penalises the model's coefficients, shrinking or setting them directly to zero, giving rise to a sparse model. Some neurons are turned off randomly, and the information is not transferred to the next layer. This technique is used to avoid overfitting. Then, the Equation (7) is transformed into

$$\mathcal{L} = \frac{1}{n} \sum_{i=1}^{n} \left( \mathbf{B}^T \mathbf{X}_i + b - y_i \right)^2 + \lambda \sum_{j=1}^{p} \left| \mathbf{B}_j \right|, \qquad (8)$$

where the last term is subject to $\sum_{j=1}^{p} \left| \mathbf{B}_j \right| < c$. The best NN architecture was also obtained by varying the model's hyperparameters, such as hidden layers between 1 and 3; the number of neurons between 32 and 512 per layer; and adjusting the learning rate across values of $10^{-2}$, $10^{-3}$, and $10^{-4}$. The best configuration has three hidden layers, with 256, 224, and 352 neurons, respectively, and a learning rate of $10^{-4}$.

## 4. Testing the Algorithms Performance

*Relative Percentage Difference*

In Figure 2, we present the relative percentage difference between the logarithm of the actual mass $M_{\mathrm{actual}}$ in the mock catalogue and the logarithm of the mass predicted by each algorithm, $M_{\mathrm{pred}}$. The algorithm dispersion is estimated by using the parameter $\Delta$ [32], which can be computed as follows

$$\Delta = 100 \times \left( \frac{\log M_{\mathrm{pred}}}{\log M_{\mathrm{actual}}} - 1 \right). \qquad (9)$$

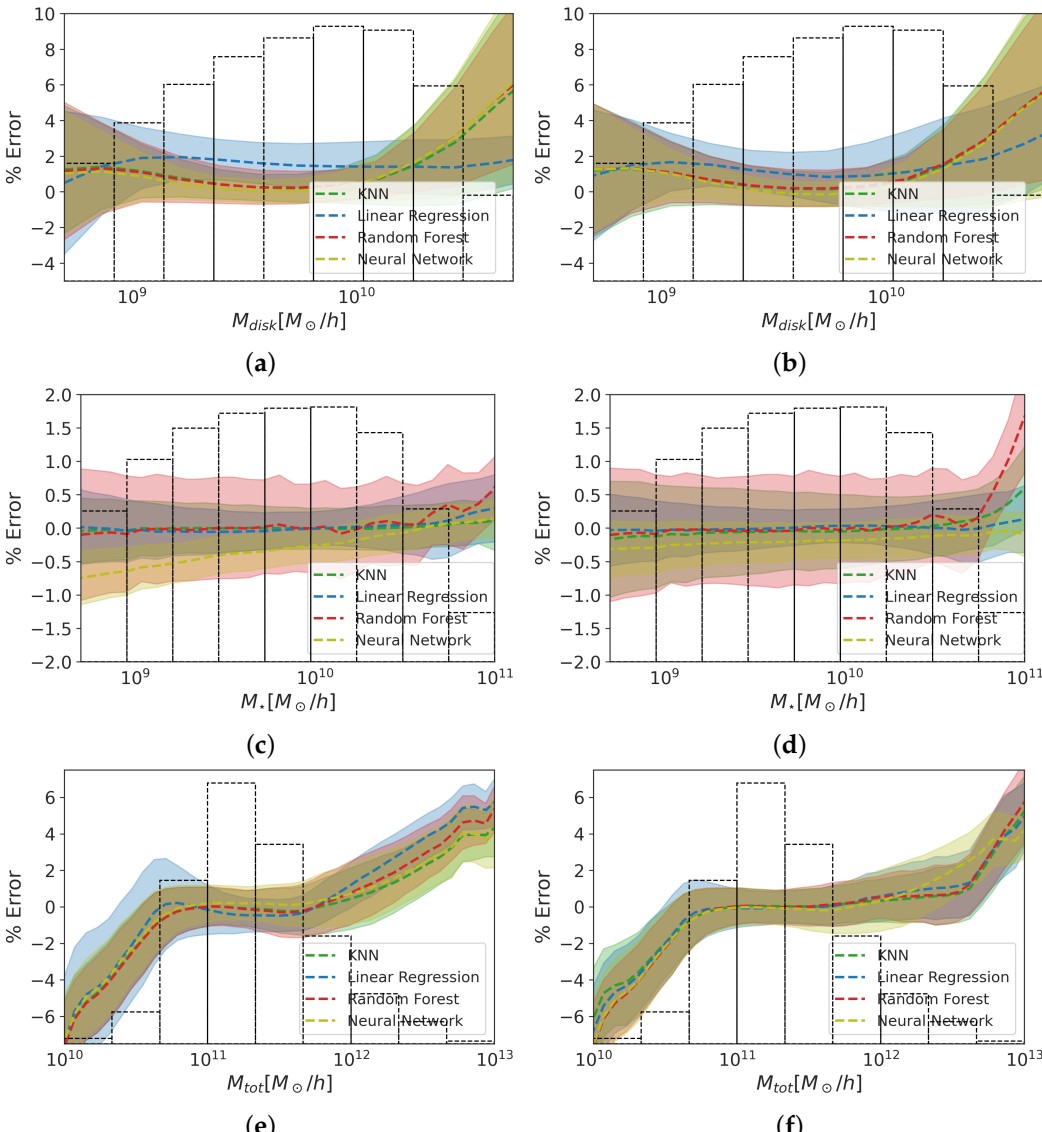

**Figure 2.** Relative percentage difference for the predictions of different ML algorithms concerning the actual values in the mock catalogues. Set I is displayed on the left, and Set II is displayed on the right. From the top to the bottom, the prediction for the output labels $M_{\text{disk}}$, $M_{\star}$ and $M_{\text{tot}}$ are shown along with different color bands associated to each algorithm (KNN in green, LR in blue, RF in red and NN in yellow green). In general, the bands follow a similar behavior and can be overlapped in certain regions. The histograms in the figures represent the distribution of the data. As expected, the predictions are better where the data density is higher. The lines represent the mean value $\mu$, and the bands are one standard deviation from the mean value $\mu \pm \sigma$.

The results were plotted into bins for which the mean value is shown in dashed lines, whereas the standard deviation corresponds to the width of the shaded regions around the mean value $\mu \pm \sigma$. The left panel shows the result when the training was carried out using Set I, while the right side corresponds to Set II.

The uncertainty bands in the histogram noticeably narrow as data counts increase, indicating a more accurate prediction. The highest errors for $M_{\text{disk}}$ and $M_{\star}$ predictions (Figure 2a,c) lie below $10^9 M_{\odot}/h$ and arise due to the low amount of data. Indeed, ML algorithms may encounter challenges in converging effectively when dealing with sparsely populated regions of the sample [33].

In contrast, for $M_{\text{tot}}$ in Figure 2e,f, the error increases for larger mass values, signifying reliable predictions in the central region around $10^{11}$–$10^{12}$, $M_{\odot}$. Notably, the distribution

of $M_{\text{tot}}$ is narrower compared with $M_{\text{disk}}$, as depicted in Figure 2a,e. This can be because the sample of galaxies chosen from the mock catalogue satisfies the condition of having a bulge, a criterion fulfilled only by sufficiently massive galaxies.

Most of the predictions exhibit statistical errors centred around zero. Figure 2c,d $M_\star$ displays the smallest percentage difference in both cases, owing to a linear correlation between magnitudes and luminous mass [34–36]. The LR model shows the best score as it was trained by directly fitting a scaling relation. In some algorithms like NN and RF, the error increases by around 1% for masses that are $10^{10} M_\odot/h$ in Set II.

In the context of disk mass, as shown in Figure 2a,b, the percentage difference is higher compared with the $M_\star$ case. Nevertheless, it remains within an acceptable prediction range for medium and high masses. Interestingly, predictions for both sets of features exhibit similar trends. In contrast to the linear fit used for $M_\star$, the LR method is no longer the optimal choice due to the nonlinear nature of this relationship. Instead, the NN and RF algorithms demonstrate superior training performance for Set I.

Finally, for $M_{\text{tot}}$, Figure 2e shows that Set I only gives unbiased predictions within the range $10.7 < \log M_{\text{tot}} < 12$, while for Set II, Figure 2f, this is true in the range $10.7 < \log M_{\text{tot}} < 12.7$. This makes sense physically as $v_{\text{max}}$ should be more sensitive for probing higher mass halos above $M_{\text{tot}} = 10^{12} M_\odot/h$. The correlation between the magnitudes in Set I and $M_{\text{tot}}$ is not straightforward. However, because mock catalogues follow the HAM relation, a correlation exists between $M_{\text{tot}}$ and $M_\star$, consequently influencing the magnitudes. This correlation contributes to achieving favourable results in predicting the total mass. In this context, NN yields the best performance for Set I, given the absence of an explicit scale relation, while for Set II, all predictions are similar.

After analysing the performance of predictions for Set I and Set II, we concluded that the latter does not significantly improve the results. As mentioned, the main enhancement is observed for $M_{\text{tot}}$. Additionally, having information about the $V_{\text{max}}$ for galaxies can be challenging due to the system's dynamics. Therefore, in the interest of simplicity, we have opted for Set I exclusively moving forward.

## 5. Predictions for Observational Data

Up to this point, we have assessed the training performance using synthetic data. In this section, we will apply the trained NN to predict masses of different components in real galaxies from the SDSS survey [20]. It is crucial to note that galaxies from the mock catalogue have specific limits for the ugriz magnitudes, which are directly tied to the resolution of the simulations. This dependence arises from the halo masses and, consequently, stellar masses, influenced by the ability of the semi-analytical models to assign magnitudes. In contrast, observed galaxies from SDSS exhibit limitations in the low surface brightness regime due to challenging observational features [37,38].

Figure 3 shows the distribution for each magnitude for both SDSS and Guo's galaxy catalogue. As previously mentioned, observed galaxies exhibit high luminosity, causing a shift in the mean value of each magnitude compared with synthetic galaxies. Because both samples do not fall within the same ranges, we will focus on regions where we have information about both observations and simulations. Indeed, the literature has reported that NN behaves as interpolators [39,40]. Therefore, the sample of observed galaxies to be assessed by the algorithm should have input features within the same domain of the training and test mock datasets.

We selected a galaxy catalogue from the SDSS database, with information about 660,000 galaxies and their morphological components [41]. The masses listed there were determined by fitting a broadband spectral energy distribution. This process involved making assumptions about the initial mass function, extinction law, and stellar evolution. In that catalogue, the bulge–disk brightness profiles were reconstructed using the photometric decomposition method with the Sersic profile

$$I(R) = I_e \exp\left\{-b_n\left[\left(\frac{R}{R_e}\right)^{1/n} - 1\right]\right\}, \tag{10}$$

where $R_e$ is the half-light radius and $I_e$ is the intensity at that radius. Here, *n* is known as the Sersic index and controls the curvature of the profile.

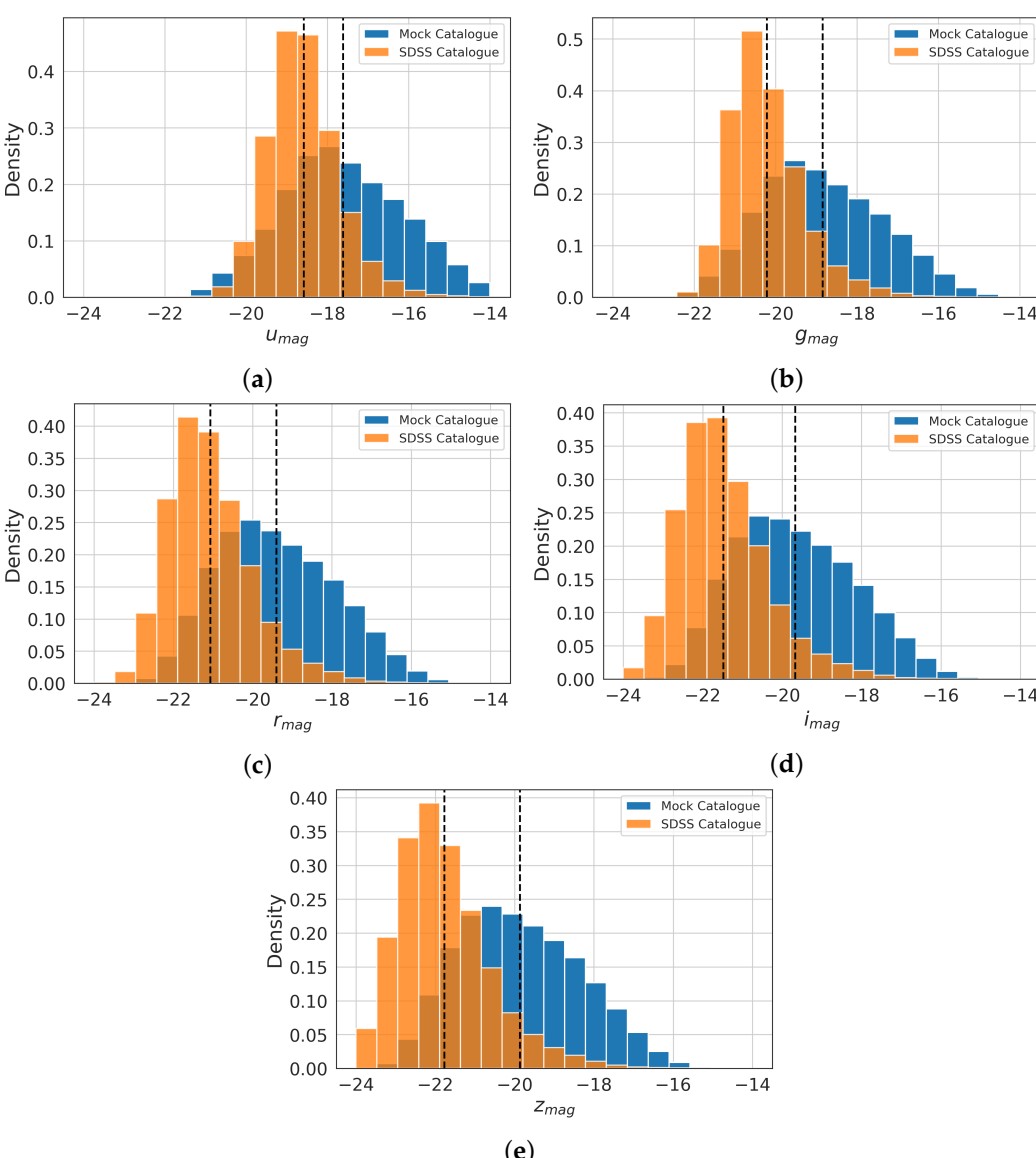

**Figure 3.** Histograms for the photometric magnitudes u, g, r, i and z are displayed in the panels (**a**–**e**), respectively. Both synthetic (in blue) and observational (in orange) catalogues have been considered. Vertical dashed lines represent the mean value of each distribution dataset. We observe that the observational case is shifted to the left in comparison with the synthetic data, in all cases. These histograms clearly show that, within the mock sample, the distribution of magnitudes for galaxies significantly differs from that for the SDSS sample. This suggests that the algorithms will explore the SDSS sample and a different combination of the features from the training set.

The magnitudes used in the prediction stage were obtained from the SDSS DR7 [20]. We converted the apparent magnitude (*m*) to absolute magnitude (*M*) using [42]

$$M = m - 5\left( \log_{10} d - 1 \right), \tag{11}$$

where *d* is the distance to the source in units of 10 parsecs. The distances were computed using the Python library Astropy [43], with the redshift reported in NED[4] and assuming the cosmological parameters from Planck 2018 [44], $H_0 = 67.66$ km/Mpc/s and $\Omega_{m0} = 0.26$.

Our analysis focused on about 70% of the total dataset, concentrating on galaxies with the u-, g-, r-, i-, and z-magnitudes.

A valuable piece of information for describing the evolution and structure of galaxies **is** the scaling relations between physical quantities of a galaxy sample. We analyse the scaling relation between mass components and the r-magnitude, as reported in other works [45–47]. This is best correlated with the stellar mass among SDSS filters. The relations for magnitudes in other colours are similar. We also study the $M_{\text{bulge}} - M_{\text{disk}}$ relation as well as the $M_\star - M_{\text{tot}}$. We only employ the NN algorithm to obtain the results presented in this section, given that it performs better with fewer errors and its construction involves a more complex architecture than the other AI algorithms.

### 5.1. Mass–Magnitude Relation

In Figure 4, we present distributions projected onto the $M_\star - r_{\text{mag}}$ and $M_{\text{disk}} - r_{\text{mag}}$ planes and the $M_{\text{bulge}} - r_{\text{mag}}$ relation for completeness. In each case, distributions up to $2\sigma$ for three datasets are shown: firstly, from the original mock catalogue in blue; then, from the original SDSS catalogue in green; and the third corresponds to NN predictions for the SDSS galaxies in red. The contours represent the 99% and 95% confidence levels. For plotting these figures, we are using the whole data of spiral galaxies in the mock catalogue; nevertheless, the masses reported in the observed catalogue fall within the regions depicted in Figure 2.

First, Figure 4 Panel (a) illustrates the scaling relation between $M_\star - r_{\text{mag}}$. The NN predictions agree with the real values up to 95% C.L. However, as we approach more massive galaxies, the resolution limit for simulations increases the error. Overall, the NN exhibits accurate predictions for $M_\star$, consistent with Figure 2c. Indeed, the best-fit slopes for each dataset only show slight differences. The best fits for the mock catalogue, SDSS, and NN predictions, respectively, are

$$M_\star = -0.427 r_{\text{mag}} + 1.370, \tag{12}$$
$$M_\star = -0.461 r_{\text{mag}} + 0.916, \tag{13}$$
$$M_\star = -0.457 r_{\text{mag}} + 0.954. \tag{14}$$

For $M_{\text{disk}}$ in Figure 4 Panel (b), we distinguish a possibly bimodal distribution with two regions for simulations laying inside the range of mass between $7 < \log M_{\text{disk}} < 9$. There is a separation between both blobs due to the lack of information at $r_{\text{mag}} \sim -19$. We report an acceptable agreement within the 95% C.L. for galaxies in the low-surface brightness region.

Here, it is worth mentioning that in all cases, the masses predicted by the NN fall within the region of the simulated masses, as expected. However, for $M_{\text{disk}}$, we observe that the red two-sigma curve moves outside the blue and green regions for masses below $10^9 M_\odot/h$ and above $5 \times 10^{10} M_\odot/h$. This is related to the fact that the output masses are distributed in a three-dimensional space (disk-bulge-stellar), and we are showing the projections over a single input parameter.

Bulge masses for most brilliant galaxies within the same mass range are not well predicted and are excluded by the NN architecture. This region corresponds to quasi-elliptical systems with large masses but small disks (see Figure 4b). The NN predicts that this type of system is unlikely, and in fact, it would be challenging to distinguish the disk from the bulge without an accurate numerical method. This conclusion is supported by Panels (a) and (c), where the prediction aligns with the expected result for more than 95% of the data. However, the missing points in Panel (b) are compensated by the excess in Panel (c). This suggests that purely elliptical systems provide a better description of these cases. This behaviour is also reflected in Figure 2a, where the error increases for masses below $10^9 M_\odot/h$.

Additionally, the fact that the neural network (NN) predicts the stellar mass of SDSS galaxies well (Figure 4a) serves as a consistency test. However, the predictions for small

disk components deviate from the SDSS catalogue, which can suggest that the features are insufficient for training the NN or that the catalogue needs further precise information about the components. The values for $M_{\text{bulge}}$ in Figure 4 Panel (c) are derived from Equation (1) and from values of $M_\star$ and $M_{\text{disk}}$ directly inferred by the NN. We can observe an acceptable agreement between observations and simulations up to a 95% C.L.

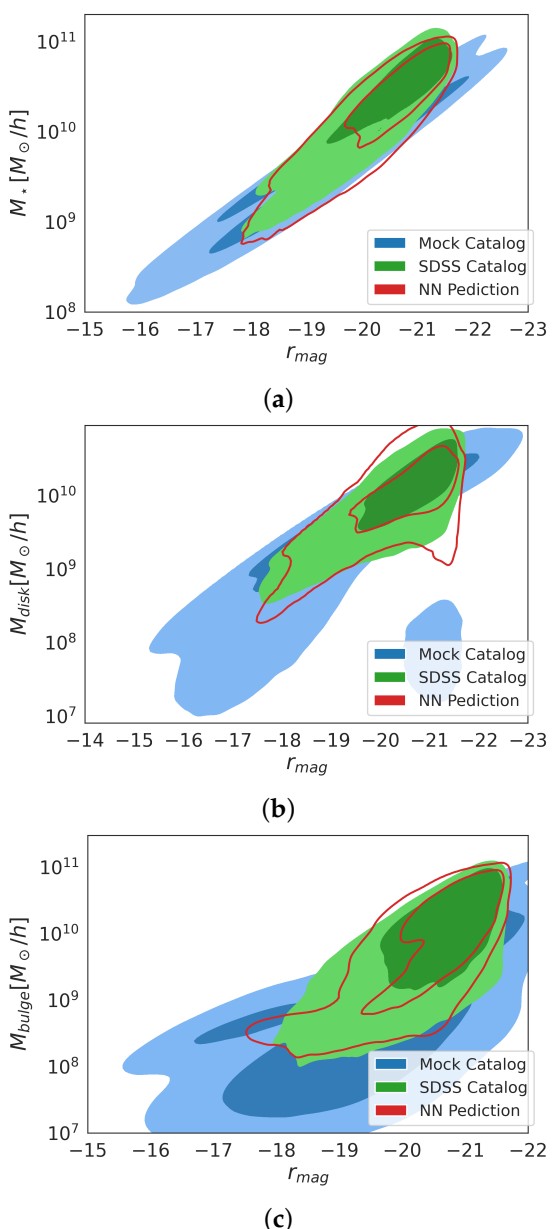

(a)

(b)

(c)

**Figure 4.** Kernel density estimation (KDE) plots of the stellar (**a**), disk (**b**), and bulge (**c**) mass components versus the r-magnitude for the simulated data in blue and the observational data in green. The red lines are the {95,99}% confidence level (C.L.) contours of the NN predictions. It can be noticed that the NN prediction is more accurate for the stellar mass and disk-dominant galaxies as the agreement is achieved up to a 99% C.L. Even though the prediction for the bulge mass is less precise than for other components, the NN archives a good agreement up to 95% C.L.

### 5.2. Bulge–Disk Components

The relation between the luminous mass and the bulge–disk masses is described by Equation (1). $M_\star$ can be determined by a scaling relation (see Figure 4). Thus, for a specific value for $M_{\text{bulge}}$, the $M_{\text{disk}}$ will only take values within certain intervals and vice versa.

Figure 5 shows the bulge and disk masses of galaxies within both datasets. The mock catalogue shows a trimodal distribution. The most prominent region for $M_{\text{disk}} > 10^8 M_{\odot/h}$ corresponds to low values for $M_{\text{bulge}}$, and it is associated with disk-dominated galaxies. The second region for $M_{\text{bulge}} > 10^{10} M_{\odot/h}$ is the bulge-dominated region [48]. These sorts of galaxies are usually dubbed as cD-like galaxies (central-dominant) [14,49]. This behaviour arises in both observed and simulated galaxies, although disk-dominant galaxies are more abundant in both cases.

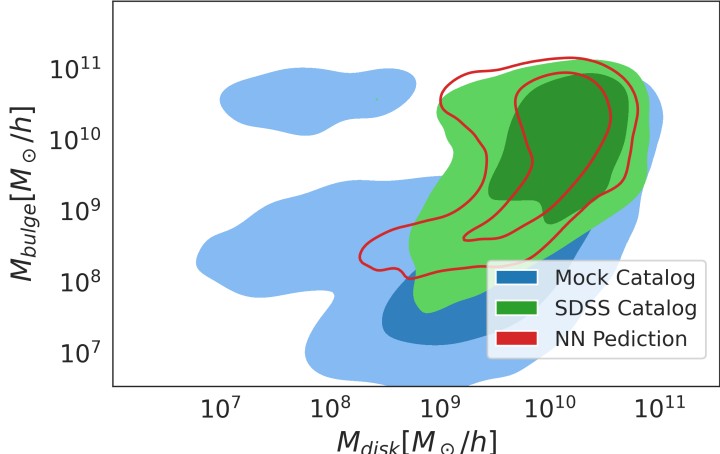

**Figure 5.** KDE plots of the bulge–disk decomposition. The distribution for simulated (blue) and observed (green) data and the solid contour levels are shown. We observe a possible trimodal distribution for the mock catalogue. In contrast, the observations show a unimodal distribution similar to the predicted NN distribution for observational data (red contours).

The third region in the $M_{\text{bulge}} - M_{\text{disk}}$ plane corresponds to galaxies with both small disks and bulges, and they are only shown for the mock data. This discrepancy suggests that there may be an observational bias because current telescopes might not be able to detect the low-luminosity galaxies that appear in the numerical simulations.

The NN prediction is also shown in Figure 5. For this last sample, the relationship between disk and bulge is nonlinear and not readily fitted with an analytic function as it happens with scaling relations derived in Section 5.1. This is related to the multimodal distribution, suggesting that different scaling relations between bulge–disk mass components might arise for different galaxies within the sample. Nevertheless, the ML algorithm can make good predictions for disk-dominant galaxies. Furthermore, it is interesting that the NN algorithm gives rise to mass predictions consistent with the SDSS distribution and does not predict bulge-dominant galaxies as expected. Giving more accurate information about larger bulges can involve more complicated dynamics.

## 6. Conclusions

It is well known that the bulge–disk decomposition and estimating the total mass of galactic systems are complicated tasks tackled using different techniques. Regarding ML algorithms, using synthetic catalogues derived from dark matter-only simulations in the training stage is common. However, such catalogues are constructed following specific prescriptions for the dark matter features as well as simplified models of the baryon dynamics, where several assumptions are usually considered [10,12]. In this sense, comparison with observations is crucial for determining the validity of the AI methods. In this work, we showed that our NN could predict the masses of the observed galaxies from the SDSS at $z = 0$ and the scaling relations between magnitudes and masses when the training was carried out using Guo's catalogue.

The NN could accurately estimate the components for disk-dominant galaxies. This also corresponds to the region where we have more information from observational data.

For this type of galaxy, only the photometric information is sufficient to separate the bulge and the disk. However, additional information is needed to make a good prediction when the bulge mass starts to dominate. This can be related to two main factors: first, $M_{\text{bulge}}$ is highly correlated with $M_{\text{bh}}$ (see Figure 1), which means that in the central region of the galaxy, the baryon dynamics play an important role. Second, the bulge-dominant galaxies' population is smaller than those with larger disks; thus, more data are necessary for both simulations and observations.

An additional feature of the NN is its capability to predict data in regions of low luminosity, where the masses associated with the bulge and the disk are small. This is particularly important in the era of precision cosmology, where we have more powerful telescopes to observe galaxies with lower luminosities than in the past. Indeed, this study can be contrasted with recent observations from the James Webb Space Telescope (JWST) or the Dark Energy Spectroscopic Instrument (DESI) [50]. At the same time, these ML tools can serve to impose constraints on the dark matter model when compared with more precise observations.

In the meantime, we will continue this project by constructing ML algorithms trained with features directly inferred from observational data to produce more accurate results in bulge-dominant galaxies. We will also explore the HAM relation in real galaxies and its possible dependency on the morphology or age of the systems using a catalogue derived from the MaNGA database, which was reported in [51]. This will also allow us to infer information about the halo mass in observed galaxies.

**Author Contributions:** Methodology, J.N.L.-S. and E.M.-V.; formal analysis, J.N.L.-S. and E.M.-V.; investigation J.N.L.-S.; writing original draft preparation,, J.N.L.-S., E.M.-V., A.A.A.-L. and O.M.M.-B.; supervision, A.A.A.-L. and O.M.M.-B. All authors have read and agreed to the published version of the manuscript.

**Funding:** The research leading to these results has received support from the European Structural and Investment Funds and the Czech Ministry of Education, Youth and Sports (project No. FORTE—CZ.02.01.01/00/22_008/0004632). This work was partially supported by Vicerrectoría de Investigación y Estudios de Posgrado of Benemérita Universidad Autónoma de Puebla under the grant 00191.

**Data Availability Statement:** Dataset available on request from the authors.

**Acknowledgments:** For this research work, we use the NASA/IPAC Extragalactic Database (NED), which is operated by the Jet Propulsion Laboratory, California Institute of Technology, under a contract with the National Aeronautics and Space Administration.

**Conflicts of Interest:** The authors declare no conflicts of interest.

## Notes

1.   In fact, when using a mock catalogue to train the algorithms, we set dark matter in a CDM prescription and make inferences about mass components holding such a hypothesis.
2.   Notice that the gas component has been neglected, as the primary light contributions in massive galaxies come from stars.
3.   Further information such as the metallicity, age, or other complex processes of baryons have not been included, as obtaining such features from observed galaxies is not straightforward.
4.   The NASA/IPAC Extragalactic Database (NED) is operated by the Jet Propulsion Laboratory, California Institute of Technology, under contract with the National Aeronautics and Space Administration.

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
