# Peer review of "Estimating the Mass of Galactic Components Using Machine Learning Algorithms"

_universe, doi:10.3390/universe10050220_

Round 1

Reviewer 1 Report

Comments and Suggestions for Authors

The paper shows interesting results on machine learning methods applied to the prediction of galactic features, such as bulge an disk mass.

The text provides adequate background about the problems tackled. The methodology is explained synthetically but in sufficient detail. Enough information is given for interested readers to potentially reproduce results, which is a good attitude. Good results are shown for a vast population of galaxies. Lack of convergence for less populated regions of the sample (such as shown in Fig. 2) is a general drawback of machine learning methods, not peculiar of this work.

I wish this study to push further, possibly with richer feature sets.

I consider the paper suitable for publication.

I spotted a few typos:

- line 145, table ?? (probably 1)

- Fig 4, "Pediction" -> "Prediction"

- line 308 "the the" -> "the"

Reviewer 2 Report

Comments and Suggestions for Authors

The Authors propose an alternative approach to predict the masses of various galactic components, such as the disk, bulge, stellar, and total mass, employing a set of machine learning algorithms: K-Nearest Neighbours, Linear Regression, Random Forest, and Neural Network (NN). This novel methodology involves using rest-frame absolute magnitudes in the ugriz-photometric system as input features, and training the models on a sample of spiral galaxies with bulges from Guo’s mock catalogue derived from the Millennium simulation.

The main finding is that all the algorithms demonstrate good predictive performance for the galaxy's mass components within the range of 10^9 to 10^11 solar masses, which corresponds to the central region of the training mass domain. However, NN exhibits the most precise predictions compared to other methods. 

To further evaluate NN architecture's performance, the Authors make use of a sample of observed galaxies from the SDSS survey with known mass components. The Authors claim that the NN can accurately predict the luminous masses of disk-dominant galaxies within the same magnitude range as the synthetic sample with up to a 99% level of confidence. 

As a bonus, the Authors report that the NN algorithm facilitates the identification of scaling relations between masses of different components and magnitudes.

The manuscript fairly meets the standards of quality, robustness and novelty for MDPI Universe.

I recommend it for publication in Universe in the present form.

Reviewer 3 Report

Comments and Suggestions for Authors

\title{\bf Review Report of Manuscript ``Estimating the mass of galactic components using machine learning algorithms"}

\maketitle

\hrulefill

\hrulefill

\begin{center}

\textbf{Comments to the Authors}

\end{center}

In the manuscript, authors presented an alternative method for predicting the masses of galactic components, 

including the disk, bulge, stellar and total mass, using a set of machine learning algorithms: KNN- 

neighbours (KNN), Linear Regression (LR), Random Forest (RF) and Neural Network (NN). The 

rest-frame absolute magnitudes in the ugriz-photometric system were selected as input features, 

and the training was performed using a sample of spiral galaxies hosting a bulge from Guo’s mock 

catalogue derived from the Millennium simulation. Study is interesting.  However, I have the following comments and suggestions before any final decision

\begin{itemize}

  \item There are some typos and grammar mistakes throughout the paper, authors need to improve it.

  \item How authors are claiming that their results indicated the physically viable. Is there any comparison?

  \item The abstract does not give any information about the main results of the research. I suggest that the authors add a few sentences to the abstract about their main findings.

  \item The conclusion section of the article is nothing but a summary of the paper. Well, how about the future of this study? or how will it guide future measurements, which have never been mentioned in the article? This discrepancy must be fixed.

      \item Some important literature is missing for example,The Astrophysical Journal, 941:170 (20pp), 2022 December 20, Fortschr. Phys. 2023, 2200129, 

\end{itemize}|

I would like to see these suggestions and criticism implemented in a revised version of this work to decide its suitability based on its novelty for publication in a good journal of the field like Universe.\\
